# Controlled LLM Training on Spectral Sphere

**Tian Xie**[1]   **Haoming Luo**[2]   **Haoyu Tang**[2]   **Yiwen Hu**[2]   **Jason Klein Liu**[3]   **Qingnan Ren**[1]   **Yang Wang**[1]
**Wayne Xin Zhao**[2,3]   **Rui Yan**[4]   **Bing Su**[2]   **Chong Luo**[1]   **Baining Guo**[1]

## Abstract

Scaling large models requires optimization strategies that ensure rapid convergence grounded in stability. Maximal Update Parametrization ($\mu$P) provides a theoretical safeguard for width-invariant $\Theta(1)$ activation control, whereas emerging optimizers like Muon are only "half-aligned" with these constraints: they control updates but allow weights to drift. To address this limitation, we introduce the **Spectral Sphere Optimizer (SSO)**, which enforces strict module-wise spectral constraints on both weights and their updates. By deriving the steepest descent direction on the spectral sphere, SSO realizes a fully $\mu$P-aligned optimization process. To enable largescale training, we implement SSO as an efficient parallel algorithm within Megatron. Through extensive pretraining on diverse architectures, including Dense 1.7B, MoE 8B-A1B, and 200-layer DeepNet models, SSO consistently outperforms AdamW and Muon. Furthermore, we observe significant practical stability benefits, including improved MoE router load balancing, suppressed outliers, and strictly bounded activations.

## 1. Introduction

LLM training is, at its core, a pursuit of **convergence speed** grounded in the necessity of **stability**. While the community has explored various optimization strategies, we argue that the essential principle governing training stability is the *Maximal Update Parametrization* ($\mu$P) (Yang et al., 2024). By mandating that the spectral norms of weights and updates scale as $\Theta(\sqrt{d_{\text{out}}/d_{\text{in}}})$ to ensure width-invariant activations remain $\Theta(1)$ scale, $\mu$P serves as the mathematical safeguard against activation explo-

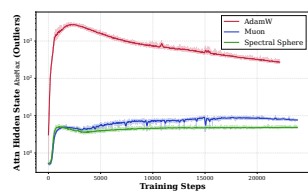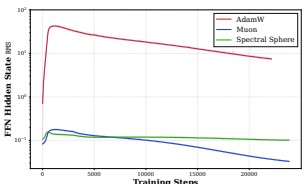

*(a)* Attn. Activations `AbsMax`      *(b)* FFN Activations `RMS`

*Figure 1.* **Training dynamics of Dense-1.7B activations (log-scaled cross-layer averages).** Our Spectral Sphere maintains constant activation RMS and empirically suppresses AbsMax outliers throughout training because of its $\mu$P-metrized constraints on the spectral manifold. Muon activations show a mild drift due to LR schedule and weight decay. By contrast, AdamW proves the most unstable, generating significantly larger activations, with attention `AbsMax` and FFN `RMS` reaching $\sim 100\times$ magnitude compared to those spectral optimizers.

sions (Takase et al., 2025). However, the current landscape is saturated with methods that fail to satisfy these fundamental conditions. Conventional soft regularization methods, such as decoupled weight decay or initialization strategies, prove insufficient over long training horizons (Kosson et al., 2025). This unconstrained weight drift destabilizes the *effective step size* (update-to-weight ratio) and degrades feature learning.

On the other side of the spectrum lies the pursuit of optimal convergence. The recent Muon optimizer (Jordan et al., 2024) has demonstrated remarkable efficiency, often interpreted as steepest descent under the spectral norm. In analyzing Muon, we uncover a surprising insight: it acts as a **"half-aligned"** solution to the $\mu$P constraints. However, maintaining stable features requires constraining not only the updates but also the weights themselves. Unstable activations like attention logits explosion were still observed in Muon training (Kimi Team et al., 2025). Consequently, practitioners are forced to rely on "non-essential" architectural patches to artificially force stability — ranging from aggressive normalization schemes like Sandwich-Norm (Ding et al., 2021) and QK-Norm (Henry et al., 2020), to ad-hoc fixes like logit softcapping (Kimi Team et al., 2025) — often at the cost of model expressivity and requiring extensive hyperparameter tuning. This observation motivates a fundamental question:

[1]Microsoft Research Asia [2]Renmin University [3]IQuest Research [4]Wuhan University. Correspondence to: Tian Xie <unakar666@gmail.com>.

*Proceedings of the 43rd International Conference on Machine Learning*, Seoul, South Korea. PMLR 306, 2026. Copyright 2026 by the author(s).

*What if an optimizer could simultaneously satisfy the steepest descent property for **convergence speed** and the strict $\mu$P constraints for **fundamental stability**?*

To answer this, we propose a mathematically unique solution that unifies these two objectives. By identifying the spectral sphere as the natural choice for stable feature learning, SSO derives the steepest descent direction constrained within this geometry. Unlike heuristic manifold projection methods (Xie et al., 2025; Pethick et al., 2025), SSO solves a constrained optimization problem in the tangent space via a Lagrange multiplier search, followed by a retraction step to map the weights back onto the spectral sphere.

To enable large-scale training, we offer a systematic implementation in Megatron. We provide principled guidelines for spectral preconditioned optimization, specifically deriving the optimal *learning rate scaler*, determining the critical *atomic module granularity*, and identifying the optimal *spectral radius* to control activation at optimal scales precisely. These offer a robust recipe for large-scale training. Specifically to mitigate the overhead of the iterative root solver, we utilize a novel distributed strategy centered on atomic module sharding (NVIDIA, 2025). This technique partitions fused params into independent spectral units, enabling communication-free local updates. We further address solver-induced workload imbalance through a size-aware ping-pong placement strategy, and accelerate matrix operations using adaptive kernel dispatcher, alongside multi-stream execution and singular vector caching.

Empirically, we validate SSO through extensive pretraining experiments across various scales, including Dense 1.7B, MoE 8B-A1B, and 200-layer DeepNet models. SSO consistently outperforms AdamW and Muon while uniquely preserving stable $\mu$P learning rate transfer. Notably, it yields superior training dynamics: SSO significantly improves MoE router load balancing, suppresses outliers in deep networks, and strictly bounds activations within a tunable scale.

## 2. Preliminary

### 2.1. Maximal Update Parametrization ($\mu$P)

$\mu$P prescribes how activations and weight updates should scale with width to preserve feature learning (Yang et al., 2024). Ideal feature learning requires the scale of activations to remain as invariant as possible. We use **operator norm** to characterize how the norm of activations changes through a linear layer.

Considering a linear layer $\boldsymbol{y} = \boldsymbol{W}\boldsymbol{x}$ with $\boldsymbol{W} \in$ $\mathbb{R}^{d_{\text{out}} \times d_{\text{in}}}, \boldsymbol{x} \in \mathbb{R}^{d_{\text{in}}}$, the RMS norm is defined as

$$\|\boldsymbol{x}\|_{\text{rms}} := \frac{\|\boldsymbol{x}\|_2}{\sqrt{d_{\text{in}}}}, \tag{1}$$

while the RMS-to-RMS operator norm is defined as

$$\|\boldsymbol{W}\|_{\text{rms}\to\text{rms}} := \sup_{\boldsymbol{x}\neq\boldsymbol{0}} \frac{\|\boldsymbol{W}\boldsymbol{x}\|_{\text{rms}}}{\|\boldsymbol{x}\|_{\text{rms}}}. \tag{2}$$

$\mu$P scale invariance requires maintaining $\|\boldsymbol{y}\|_{\text{rms}} = \|\boldsymbol{x}\|_{\text{rms}} = \Theta(1)$, which is equivalent to enforcing the RMS-to-RMS condition $\|\boldsymbol{W}\|_{\text{rms}\to\text{rms}} = \|\boldsymbol{W}\|_2 \sqrt{d_{\text{in}}/d_{\text{out}}} = \Theta(1)$. This induces the following spectral norm constraint on the weight matrix $\|\boldsymbol{W}\|_2 = \Theta(\sqrt{d_{\text{out}}/d_{\text{in}}})$. A similar requirement applies to parameter updates; together, we refer to these as the spectral $\mu$P condition below.

**Remark 2.1** (Spectral $\mu$P Condition). Yang et al. (2024) show that preserving *scale-invariant activations* for feature learning requires the same law to hold for both weights and their updates:

$$\|\boldsymbol{W}\|_2 = \Theta\left(\sqrt{\frac{d_{\text{out}}}{d_{\text{in}}}}\right), \quad \|\boldsymbol{\Phi}\|_2 = \Theta\left(\sqrt{\frac{d_{\text{out}}}{d_{\text{in}}}}\right).$$

### 2.2. Steepest Descent under Different Norms

Following Bernstein & Newhouse (2024); Franz Louis Cesista (2025), many successful optimizers can be interpreted as **first-order** methods without convexity assumptions. Specifically, after switching off exponential moving averages, these algorithms reduce to instances of steepest descent governed by distinct norms: SGD corresponds to the Frobenius norm $\|\cdot\|_F$, AdamW (Loshchilov & Hutter, 2019) to the $\ell_\infty$ norm, and Shampoo (Gupta et al., 2018) to the spectral norm $\|\cdot\|_2$.

In this view, we determine the update $\boldsymbol{\Phi}$ by iteratively minimizing a first-order Taylor approximation of the loss around the current weights $\boldsymbol{W}$, subject to a hard constraint on the step size:

$$\boldsymbol{\Phi} := \operatorname*{argmin}_{\|\boldsymbol{\Phi}\|\leq\eta} \{\mathcal{L}(\boldsymbol{W}) + \langle\boldsymbol{G}, \boldsymbol{\Phi}\rangle\} = \operatorname*{argmin}_{\|\boldsymbol{\Phi}\|\leq\eta} \langle\boldsymbol{G}, \boldsymbol{\Phi}\rangle, \tag{3}$$

where $\boldsymbol{G} := \nabla_{\boldsymbol{W}}\mathcal{L}(\boldsymbol{W})$ is the gradient of the loss function at $\boldsymbol{W}$. The update is thus determined by two priors: a **norm** $\|\cdot\|$ assigned according to the specific *functional role* of the module, and a **step size** $\eta$ that governs the update scale. The solution is as follows:

**Definition 2.2** (Steepest Descent Update). Given a norm $\|\cdot\|$ that endows the parameter space with a geometry, the **steepest descent update** is

$$\boldsymbol{\Phi} = -\eta \cdot \boldsymbol{\Phi}_{\text{unit}}, \quad \text{where} \quad \boldsymbol{\Phi}_{\text{unit}} := \operatorname*{argmax}_{\|\boldsymbol{T}\|\leq 1} \langle\boldsymbol{G}, \boldsymbol{T}\rangle. \tag{4}$$

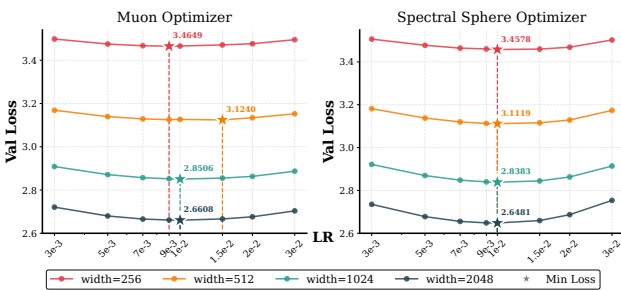

*Figure 2.* **$\mu$P width scaling across $25\times$ model sizes (70M to 1.8B).** Although $\mu$P aims for width-invariant scaling, Muon still exhibits notable optimal learning rate drift. In contrast, our Spectral Sphere achieves stable LR transfer, while also obtaining **lower optimal loss than** Muon. More details and related experiments are provided in Appendix A.5.

### 2.3. Muon Optimizer

Following the framework of metrized deep learning (Section 2.2), Muon (Jordan et al., 2024) can be interpreted as steepest descent under the **spectral norm**. For a 2D weight matrix $W$, the spectral norm $\|\cdot\|_2$ [1] gives the tightest bound on the matrix's input-output gain:

$$\|W\|_2 := \sup_{x \neq 0} \frac{\|Wx\|_2}{\|x\|_2}, \qquad (5)$$

By choosing the spectral norm to constrain the update direction $\Phi$, the steepest descent direction is uniquely given by the **matrix sign function (msign)** (Appendix A.1):

$$\mathrm{msign}(G) = U\,\mathrm{sign}(\Sigma)V^\top = U_{[:,:r]}V_{[:,:r]}^\top, \qquad (6)$$

where $G = U\Sigma V^\top$ is the singular value decomposition (SVD). The msign operation orthogonalizes the gradient, equalizing all active singular directions and yielding a spectrum isotropic update. A key contribution of Muon is the efficient approximation of $\mathrm{msign}(G)$ via Newton–Schulz iterations on GPUs.

However, Muon constrains only the backward update $\Phi$, leaving the forward weights $W$ unconstrained. This often leads to unstable $\mu$P feature learning in hidden states rms, motivating our development of **a fully aligned approach that constrains both $W$ and $\Phi$**.

## 3. Method

### 3.1. Optimization Target Formulation

We start by focusing on a hidden layer matrix $W \in \mathbb{R}^{d_{\mathrm{out}} \times d_{\mathrm{in}}}$. To satisfy the spectral $\mu$P scaling condition in Section 2.1, we set the spectral scale to target radius $R$:

$$R = \Theta\left(\sqrt{d_{\mathrm{out}}/d_{\mathrm{in}}}\right). \qquad (7)$$

Following metrized deep learning (Section 2.2), we define the unit update $\Phi$ as the solution to a constrained optimization problem:

**Formulation 3.1** (Steepest Descent on the Spectral Sphere)**.** We perform **steepest descent** under the **spectral norm**, constraining both the **weights** and the **updates** to a spectral sphere of radius $R$. Specifically, we parameterize the update step as $\Delta W = \eta R \Phi$, where $\eta$ is the base learning rate. The update direction $\Phi$ is the solution to

$$\begin{aligned} \max_{\Phi} \quad & \langle G, \Phi \rangle \\ \text{s.t.} \quad & \|\Phi\|_2 = 1, \\ & \|W - \eta R \Phi\|_2 = \|W\|_2 = R. \end{aligned} \qquad (8)$$

### 3.2. First-Order Tangent Space Constraint

Assuming the top singular value is unique[2], the spectral norm $\|W\|_2$ is differentiable with gradient $\Theta := \nabla_W \|W\|_2 = u_1 v_1^\top$, where $(u_1, v_1)$ are the principal left and right singular vectors (Watson, 1992)[3]. We consider first-order Taylor Expansion of spectral norm around $W$

$$\|W - \eta R \Phi\|_2 = \|W\|_2 - \eta R \langle \Theta, \Phi \rangle + \mathcal{O}(\eta^2 R^2 \|\Phi\|_2^2). \qquad (9)$$

To enforce the invariance condition $\|W - \eta R \Phi\|_2 = \|W\|_2$, the first-order term must vanish, which implies the tangent constraint: $\langle \Theta, \Phi \rangle = 0$. Equation (8) thus reduces to

$$\max_{\Phi} \langle G, \Phi \rangle \quad \text{s.t.} \quad \|\Phi\|_2 = 1, \ \langle \Theta, \Phi \rangle = 0. \qquad (10)$$

We then introduce a *Lagrange multiplier* $\lambda$ and maximize *Lagrangian* $\mathcal{L}(\Phi; \lambda) = \langle G + \lambda\Theta, \Phi \rangle$ under constraint $\|\Phi\|_2 = 1$. The analytical solution and numerical method are summarized below.

**Theorem 3.2.** *For fixed $\lambda$, the steepest descent direction is*

$$\Phi^\star(\lambda) = \mathrm{msign}(G + \lambda\Theta), \qquad (11)$$

*where $\lambda^\star$ is the unique root of the constraint function*

$$h(\lambda) := \langle \Theta, \mathrm{msign}(G + \lambda\Theta) \rangle, \quad h(\lambda^\star) = 0. \qquad (12)$$

*Proof.* Proof is provided in Appendix A.1. □

With proof in Appendix A.2 and visualization in Figure 3, $h(\lambda)$ has below theoretical properties.

---

[1] For vectors, $\|v\|_2$ denotes the $\ell_2$ norm; for matrices, $\|W\|_2$ denotes the (induced $\ell_2 \to \ell_2$) spectral norm in our paper.

[2] Numerical coincidence of singular values is a measure-zero event for random matrices (Tao & Vu, 2014). Quantitatively, it occurs with probability at most $\exp(-c\max(d_{\mathrm{in}}, d_{\mathrm{out}}))$ (Han, 2025).

[3] In rare degenerate cases with multiplicity, $\Theta$ serves as a sub-gradient ($\Theta \in \partial\|W\|_2$).

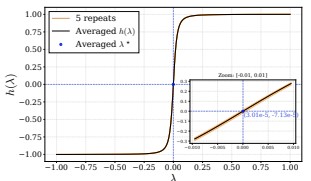
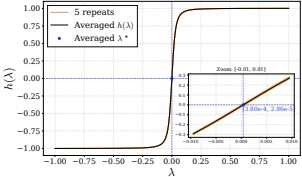

*(a)* $1024 \times 3072$ matrix     *(b)* $4096 \times 1024$ matrix

*Figure 3.* **Empirical curves of** $h(\lambda) = \langle \boldsymbol{\Theta}, \text{msign}(\boldsymbol{G} + \lambda\boldsymbol{\Theta})\rangle$ **for random matrices**. $h(\lambda)$ is monotonic non-decreasing in $\lambda$, and its root $\lambda^\star$ lies close to zero (proof in Appendix A.2). Here, each curve is obtained by averaging over 5 repeats, and each matrix is initialized from $\mathcal{N}(0, 0.02^2)$.

i) **Monotonicity:** The function $h(\lambda)$ is monotonically non-decreasing and transitions from $-1$ to $+1$ as $\lambda$ varies from $-\infty$ to $+\infty$.

ii) **Root Localization:** The solution $\lambda^\star$ is strictly bounded within the interval $[-2\|\boldsymbol{G}\|_*, 2\|\boldsymbol{G}\|_*]$, providing a finite search space [4].

With the properties above, we can derive Lemma 3.3 while the overhead analysis is provided in Section 5.

**Lemma 3.3.** *Given the monotonic nature of* $h(\lambda)$*, we locate* $\lambda^\star$ *efficiently:*

▷ *Bracketing: Leveraging monotonicity, we start from* $\lambda = 0$ *and exponentially expand the search bracket in the opposite direction of the sign of* $h(0)$ *until the root is enclosed.*

▷ *Bisection: We isolate* $\lambda^\star$ *via standard bisection within the bracketed interval.*

### 3.3. Second-Order Manifold Constraint

Note that the remainder $\mathcal{O}(\eta^2 R^2 \|\boldsymbol{\Phi}\|_2^2)$ in Equation (9) may accumulate over iterations, causing gradual drift off the spectral sphere. To enforce the exact constraint $\|\boldsymbol{W}\|_2 = R$ throughout training, we apply a retraction step that projects the weights back onto the manifold:

$$\boldsymbol{W} \leftarrow \boldsymbol{W} \cdot \frac{R}{\|\boldsymbol{W}\|_2}. \tag{13}$$

While the retraction is conceptually a post-update projection, we implement it as a pre-update correction for efficiency, which is operationally equivalent. This reordering allows us to invoke the computationally expensive Power Iteration only once per step, reusing the resulting singular triplet for both the manifold retraction and the tangent projector $\boldsymbol{\Theta}$ (Lines 6–9 in Algorithm 1).

The retraction strictly constrains $\|\boldsymbol{W}\|_2 = R$, which automatically bounds the Frobenius norm $\|\boldsymbol{W}\|_F$ (as shown

---

[4]Nuclear norm $\|\cdot\|_*$ is the sum of singular values.

---

**Algorithm 1** Spectral Sphere Optimizer (SSO)

---

**Require:** Initial 2D weights $\boldsymbol{W}_0 \in \mathbb{R}^{d_{\text{out}} \times d_{\text{in}}}$, spectral $\boldsymbol{\mu}$P scaler $R = \sqrt{d_{\text{out}}/d_{\text{in}}}$, learning rate $\eta$, momentum coefficient $\beta$, precision tolerance $\epsilon$

1: **Initialize:** $\boldsymbol{W}_0 \leftarrow R \cdot \boldsymbol{W}_0 / \|\boldsymbol{W}_0\|_2$, $\boldsymbol{M}_0 \leftarrow 0$
2: **for** $t = 0, 1, \ldots$ **do**
3:    $\boldsymbol{G}_t \leftarrow \nabla_{\boldsymbol{W}} \mathcal{L}(\boldsymbol{W}_t)$
4:    $\boldsymbol{M}_t \leftarrow \beta\boldsymbol{M}_t + (1 - \beta)\boldsymbol{G}_t$
5:    $\widehat{\boldsymbol{M}}_t \leftarrow \boldsymbol{M}_t / \|\boldsymbol{M}_t\|_F$
            ▷ Normalize for Numerical Stability
   *// 1. Spectral Geometry Analysis*
6:    $(\sigma_t, \boldsymbol{u}_t, \boldsymbol{v}_t) \leftarrow \text{PowerIteration}(\boldsymbol{W}_t)$
            ▷ Top Singular Value & Vectors
7:    $\boldsymbol{\Theta}_t \leftarrow \boldsymbol{u}_t \boldsymbol{v}_t^\top$
            ▷ Tangent Space Projector
   *// 2. Retraction to Spectral Sphere*
8:    $\boldsymbol{W}_t \leftarrow \boldsymbol{W}_t \cdot R/\sigma_t$
   *// 3. Steepest Descent Lagrange Solver*
9:    Define $h(\lambda) := \langle \boldsymbol{\Theta}_t, \text{msign}(\widehat{\boldsymbol{M}}_t + \lambda\boldsymbol{\Theta}_t)\rangle$
10:   $\lambda_t^* \leftarrow \text{Bisection}(h, \text{tolerance} = \epsilon)$
            ▷ Find root of $h(\lambda) = 0$
   *// 4. $\mu$P-Scaled Update*
11:   $\boldsymbol{\Phi}_t \leftarrow \text{msign}(\widehat{\boldsymbol{M}}_t + \lambda_t^* \boldsymbol{\Theta}_t)$
12:   $\boldsymbol{W}_{t+1} \leftarrow \boldsymbol{W}_t - \eta \cdot R \cdot \boldsymbol{\Phi}_t$
            ▷ $\boldsymbol{\mu}$P Style Update
13: **end for**

---

below). As a result, weight decay, primarily introduced to limit the weight scale, becomes redundant. We therefore **eliminate weight decay** in hidden 2D weights[5], replacing its implicit regularization with an explicit target radius $R$. Details can be found in Appendix A.3.

$$\|\boldsymbol{W}\|_F \leq \sqrt{\text{rank}(\boldsymbol{W})}\,\|\boldsymbol{W}\|_2 \leq \sqrt{\min(d_{\text{out}}, d_{\text{in}})}\,R.$$

In this paper, we set AdamW and Muon as baselines. Additionally, we introduce MuonSphere, a variant similar to Scion (Pethick et al., 2025)[6], which can be viewed as Spectral Sphere with $\lambda = 0$. While Spectral Sphere follows the exact steepest-descent approach, MuonSphere simply normalizes 2D hidden weights onto the spectral sphere before each update, deviating from the optimal trajectory. Figure 4 provides a sketch of the latter three spectral-preconditioned optimizers' update.

---

[5]We apply weight decay to params like `Embedding` for possible scaling stability, although our ablations on 1.7B model suggest that disabling weight decay may actually be slightly better.

[6]Unlike Scion, which applies ColNorm→Spectral→Sign ($l_\infty$) norm chain throughout the network, MuonSphere retains Sign→Spectral→Sign norm scheme. We find ColNorm input hurts performance.

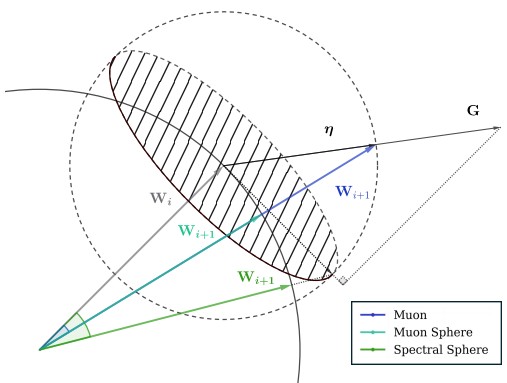

*Figure 4.* **Geometry of Steepest Descent Update Directions.** The left solid arc denotes the $W$ sphere, while the right dotted arc denotes the $\Delta W$ sphere (unit $\Phi$ scaled by $\eta$). The shaded region represents the feasible set within the *tangent space* of the $W$ sphere at step $W_i$. Under weight constraint, projecting $G$ onto the tangent space (Spectral Sphere) yields largest update angle.

## 4. Algorithm Details

### 4.1. Spectral Radius Scale

Given the target spectral radius

$$R = \Theta\left(\sqrt{d_{\text{out}}/d_{\text{in}}}\right) = c\left(\sqrt{d_{\text{out}}/d_{\text{in}}}\right), \quad (14)$$

the constant $c$ serves as a scalar that sets the branch output magnitude relative to the residual stream. By tuning $c$, one can precisely control the signal-to-noise ratio along the deep residual path, balancing the contributions of the Attention/FFN blocks against the skip connection. Properly choosing $c$ is therefore essential for stabilizing depth-wise signal propagation in Transformers. Ablation results are provided in Figure 5.

### 4.2. Learning Rate Scaler

We propose a unified view in which each learning rate scaler enforces a consistent **effective step size** under a chosen **norm metric** and **initialization scheme**.

In the update rule $W \leftarrow W - \eta R\Phi$, $R$ scales the update size. To avoid instability caused by heterogeneous layer shapes, we select $R$ to maintain a constant *effective step size* — defined as the ratio of update-to-weight magnitude under a norm metric $\|\cdot\|$:

$$\frac{\|\Delta W\|}{\|W\|} = \frac{\|\eta R\Phi\|}{\|W\|} \approx \eta. \quad (15)$$

We evaluate three common learning rate scalers below.

$$R = \begin{cases} \sqrt{d_{\text{out}}/d_{\text{in}}}, & \text{(Spectral } \boldsymbol{\mu}\text{P)} \\ \sqrt{\max(d_{\text{out}}, d_{\text{in}})} \cdot 0.2, & \text{(Align-Adam-RMS)} \\ \sqrt{\max(1, d_{\text{out}}/d_{\text{in}})}, & \text{(Spectral Kaiming)} \end{cases}$$

- **Spectral $\boldsymbol{\mu}$P Scaler.** Enforces the *RMS-to-RMS operator norm* invariance from Section 2.1. It ensures that both $W$ and $W + \Delta W$ satisfy the $\boldsymbol{\mu}$P scaling $\|W\|_2 = \Theta(\sqrt{d_{\text{out}}/d_{\text{in}}})$, forming the geometric basis of our Spectral Sphere Optimizer.

- **Align-Adam-RMS Scaler.** A heuristic for consistent relative learning rates in the *RMS norm* under fixed standard-deviation initialization. Empirically, it aligns per-layer update RMSnorm to AdamW, enabling direct transfer of AdamW-tuned hyperparameters (e.g. learning rate, weight decay) to the spectral method (Liu et al., 2025).

- **Spectral Kaiming Scaler.** Targets the *spectral norm* under Kaiming initialization ($W \sim \mathcal{N}(0, 1/d_{\text{in}})$) (He et al., 2015). Random matrix theory establishes that such matrices satisfy $\|W\|_2 \approx 1 + \sqrt{d_{\text{out}}/d_{\text{in}}}$. This scaling prevents vanishing pre-activations in bottleneck layers where $d_{\text{out}} \ll d_{\text{in}}$ (Pethick et al., 2025).

Experimental results (Figure 6a) favor the **Spectral $\boldsymbol{\mu}$P** scaler, supporting the theoretical scaling condition in Section 2.1. This demonstrates that *optimizing on a spectral manifold requires a scaler explicitly calibrated to the spectral norm.*

### 4.3. Module Granularity

To maximize computational efficiency, modern Transformer implementations such as Megatron-LM (Shoeybi et al., 2019) fuse several matrices into a single physical tensor (e.g. the QKV projections or the SwiGLU gate/up matrices). However, since these modules have distinct functional roles, imposing a unified constraint on the fused tensor is suboptimal as shown in Figure 6b. Instead, we decompose fused tensors and treat each submatrix as an independent module. We apply **per module spectral initialization and optimization** at this finer granularity by default (e.g. splitting attention QKV per head and separating FFN gate/up). Note that granularity is tunable depending on infra speed.

Under our spectral $\boldsymbol{\mu}$P initialization scheme (also discussed in Section 4.2), the weight matrix is constructed by first sampling $W_k \sim \mathcal{N}(0, \sigma^2)$ and subsequently projecting it onto the spectral sphere:

$$W = c\sqrt{\frac{d_{\text{out}}}{d_{\text{in}}}} \cdot \frac{W_k}{\|W_k\|_2}, \quad (16)$$

where $c$ denotes the spectral radius scale in Section 4.1.

## 5. Infrastructure Design

The main challenge in **SSO** implementation comes from the **bracket-and-bisect** root solver that runs at every up-

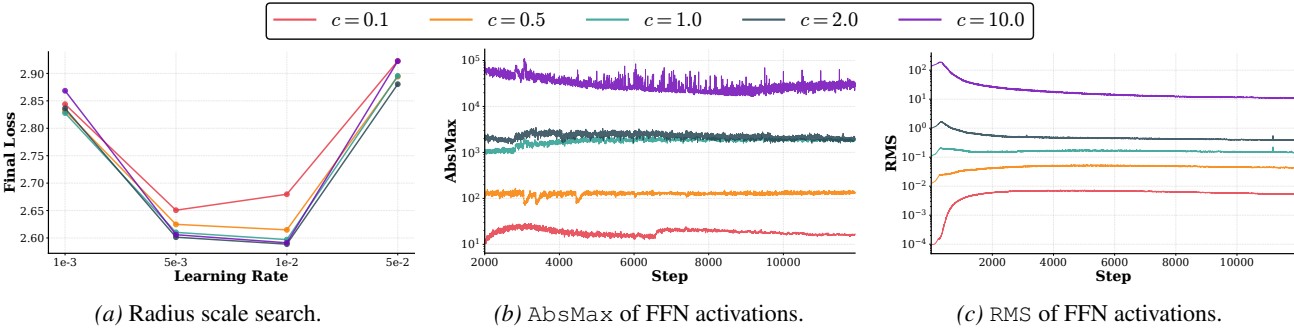

*(a)* Radius scale search.      *(b)* `AbsMax` of FFN activations.      *(c)* `RMS` of FFN activations.

*Figure 5.* **Ablation of radius scaling on *optimization* and *activation*.** (a) Final loss vs. learning rate for varying radius scales $c$. A moderate scale (e.g. $c = 2.0$) achieves the best performance. (b) AbsMax and (c) RMS of FFN activations during training. AbsMax monotonically follows the radius scale, whereas RMS follows a clear power-law scaling with $c$.

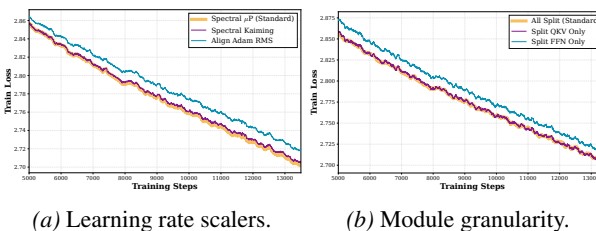

*(a)* Learning rate scalers.      *(b)* Module granularity.

*Figure 6.* **Ablation studies.** (a) Learning rate scalers: each curve shows the optimal validation loss from a learning rate grid search. Spectral $\mu$P outperforms Align-Adam-RMS, validating $\mu$P-aligned scaling under the Spectral $\mu$P condition. (b) Initialization and optimization module granularity: splitting QKV per head alone yields the most significant performance gain. Although splitting the FFN gate/up weights produces nearly the same loss curve as no-split, we maintain this split to respect their distinct functional roles; the overlapping curve is omitted from the figure.

date. To satisfy the tangent-space constraint, we must find a Lagrange multiplier $\lambda$ such that $h(\lambda) = 0$. We first expand the search interval by **bracketing**, and then we run **bisection** inside the bracket to obtain a $\lambda^\star$ that meets a target tolerance. This solver introduces non-trivial overhead:

1. **Workload Imbalance:** different bracketing range and tolerance settings can change the number of search and bisection steps, leading to unstable runtime and workload imbalance between devices.

2. **Computational Cost:** Each step in the search also evaluates $h(\lambda)$, which calls $\mathrm{msign}(\widehat{M} + \lambda\Theta)$; these extra matrix computations accumulate and add noticeable cost to every optimizer step.

3. **Synchronization Overhead:** Iterative search creates frequent synchronization between the GPU and the CPU, because the algorithm must finish each evaluation and then check a scalar condition before choosing the next $\lambda$ and continuing.

### 5.1. Optimization Pipeline

We introduce a holistic optimization pipeline designed to mitigate these overheads while preserving numerical precision. **All performance statistics are collected from Dense 1.7B model pretraining.**

**Atomic Module Sharding.** We employ a fine-grained, parameter-wise sharding strategy (NVIDIA, 2025). As noted by (Liu et al., 2025), while standard ZeRO-1 is efficient for element-wise optimizers (e.g. AdamW), its flat-buffer sharding approach is incompatible with spectral methods that require full gradient matrices to compute updates. To reconcile this, we shard parameters as *atomic modules* rather than flattened buffers. An atomic module is defined as the minimal independent weight matrix required to remain intact for spectral operations (see Section 4.3).

**Load Balancing Strategy.** To resolve the workload imbalance caused by variable solver depths, we employ a "ping-pong" load balancing strategy adapted from (NVIDIA, 2025). Empirical results indicate this method outperforms both greedy size-descent sorting and default round-robin allocation. We sort atomic modules by size and assign them to DP ranks in an alternating zigzag pattern. This heuristic effectively balances the solver workload without complex runtime scheduling. Finally, we synchronize the updated params using iterative All-Gather collective from (NVIDIA, 2025).

**Adaptive Kernel Selection.** We observe that the optimal implementation for Matrix Sign computation is highly sensitive to matrix dimensions. As shown in Table 1, applying specialized kernels indiscriminately can degrade performance. We therefore implement an Adaptive Dispatcher:

- **Small Matrices** ($< 512$): We use a JIT-compiled PyTorch implementation built on `torch.addmm`. This avoids the launch overhead of specialized kernels.

*Table 1.* Optimization breakdown on end-to-end latency for 4M tokens/step on NVIDIA B200. ↓ denotes improvement, while ↑ denotes regression. Note there is still room for improvement.

|  | Time (ms) | Δ vs Baseline | Δ vs Prev. |
|---|---|---|---|
| Naive Baseline (No opt.) | 10928.5 | - | - |
| + Load balance & All Gather | 9365.5 | -1563.0 (-14.3%)↓ | -1563.0 (-14.3%)↓ |
| + Triton SYRK Kernel | 10284.2 | -644.3 (-5.9%)↓ | +918.7 (+9.8%)↑ |
| + Adaptive & Multi-stream | 9383.4 | -1545.1 (-14.2%)↓ | -900.8 (-8.8%)↓ |
| + BF16 & Torch.compile (**Final**) | **7666.3** | **-3262.2 (-29.9%)↓** | **-1717.1 (-18.3%)↓** |

*Table 2.* End-to-end per-step latency for 4M tokens/step on NVIDIA B200. Muon as baseline.

|  | AdamW | Muon | MuonSphere | Spectral Sphere |
|---|---|---|---|---|
| Time (ms) | 6734.15 (-2.10%) | 6878.83 | 6949.85 (+1.03%) | 7666.32 (+11.45%) |

- **Large Matrices ($\geq$ 512)**: We dispatch to a custom Triton kernel implementing the SYmmetric Rank-K (SYRK) (NVIDIA, 2025) update, which exploits the symmetry of Newton–Schulz iterations to halve memory reads.

**Multi-Stream Parallelism.** For layers composed of many small independent matrices (e.g. per head attention split), single-stream execution suffers from kernel launch latency bubbles. We exploit this independence by dispatching spectral updates across multiple CUDA streams.

**Mixed-Precision.** The Power Iteration for spectral norm estimation is performed in BFloat16, while the sensitive **msign remains in FP32 with 8 iterations**.

**Cache Top Singular Vectors.** The singular vectors of model weights evolve slowly during training. Leveraging this temporal locality, we initialize the current Power Iteration using the cached singular vectors $u$ and $v$ from the previous step. This reuse mechanism drastically accelerates convergence, requiring only a few iterations to maintain approximation accuracy.

## 6. Scaling Experiments

### 6.1. Experimental Setup

**Hyperparameters.** Following the $\mu$P protocol, we perform a learning rate sweep on the 1.7B model with AdamW between $[10^{-3}, 10^{-2}]$ and find $5 \times 10^{-3}$ to be the optimal value. Training uses 500 warmup steps, a global batch size of $\sim$4M tokens (1024 sequences $\times$ 4096 tokens each), and a cosine learning rate decay reduced to $10\%$ peak. We use BF16 mixed-precision training.

Following Kimi Moonlight (Liu et al., 2025), all optimizers use a weight decay of 0.1. However, distinct from their approach of aligning updates to AdamW RMS (intended to reuse current scaling laws), we find that spectral $\mu$P LR

scaler outperforms the uniform 0.2 update-rms alignment (Section 4.2). For msign coefficients, we use the Polar Express method (Amsel et al., 2025) with 8 NewtonSchulz iterations[7]. We employ Nesterov momentum and, by default, split attention heads and FFN gate/up projections for separate initialization and optimization. For Spectral Sphere, we set the $\lambda$-solver's maximum iterations to 20, Lagrange solver precision tolerance to 2e-4, and remove weight decay for all hidden 2D weights, as retraction maintains the weight constraint (Section 3.3).

**Training Data.** We use the OLMo-Mix-1124 dataset (Team OLMo et al., 2025), tokenized with the OLMo-2 tokenizer. We randomly sample 100 billion tokens for training and reserve 1 billion tokens for validation. Data indices are built offline to guarantee a deterministic training order.

**Benchmarks.** We primarily evaluate on the following downstream tasks: ARC (Clark et al., 2018), BoolQ (Clark et al., 2019), CSQA (Talmor et al., 2019), HellaSwag (Zellers et al., 2019), PIQA (Bisk et al., 2020), WinoGrande (Sakaguchi et al., 2021) and LAMBADA (Paperno et al., 2016).

### 6.2. Dense 1.7B

We adopt the architecture configuration of Qwen3-1.7B (Qwen-Team et al., 2025), replacing its original tokenizer with that of OLMo-2 (Team OLMo et al., 2025). The core architecture utilizes Grouped Query Attention (GQA), QK-Norm, SwiGLU activations, Rotary Positional Embeddings (RoPE), and pre-normalization RMSNorm. We do not tie the embedding and the head.

Figure 7 shows the validation loss of different optimizers. Spectral Sphere shows $1.24\times$ speedup compared to AdamW. Results in Table 3 show that the two Sphere optimizers outperform the baselines on the common-sense reasoning tasks, with Spectral Sphere the best on average.

### 6.3. MoE 8B-A1B

The configuration largely follows DeepSeek-V3 (DeepSeek-AI et al., 2025). The model has 27 layers: the first is a standard dense FFN, followed by 26 MoE layers. We utilize 64 experts in total, with a top-4 routing expert plus 1 shared expert.

**Router and Load Balancing.** The router is implemented in FP32 precision. We adopt the auxiliary-loss-free strategy (Wang et al., 2024), which demonstrates superior ex-

---

[7]We tested 5 and 8 iterations as well as different msign coefficients and observed negligible differences in training loss ($<$1e-3). We retain 8 iterations for improved numerical accuracy.

*Table 3.* **Performance comparison of different optimizers on Dense 1.7B models. Bolded** indicate the best among different settings.

| Optimizer | LMB. (PPL) ↓ | LMB. (Acc) ↑ | CSQA (Acc) ↑ | PIQA (Acc) ↑ | Hella. (Acc) ↑ | Wino. (Acc) ↑ | ARC-e (Acc) ↑ | ARC-c (Acc) ↑ | BoolQ (Acc) ↑ | Avg. (Acc) ↑ |
|---|---|---|---|---|---|---|---|---|---|---|
| AdamW | 5.40 | 63.71 | 19.66 | 74.70 | 47.90 | 62.59 | 68.81 | 37.37 | 63.24 | 54.75 |
| Muon | 5.05 | 65.19 | 19.00 | 75.35 | 48.91 | 61.72 | 70.24 | 37.46 | 64.22 | 55.26 |
| MuonSphere | **4.87** | **65.55** | 20.07 | 74.97 | 49.20 | 62.83 | 71.51 | **38.40** | **66.97** | 56.19 |
| Spectral Sphere | 5.00 | 65.07 | **21.05** | **75.95** | **49.25** | **63.77** | **71.80** | 38.31 | 65.57 | **56.35** |

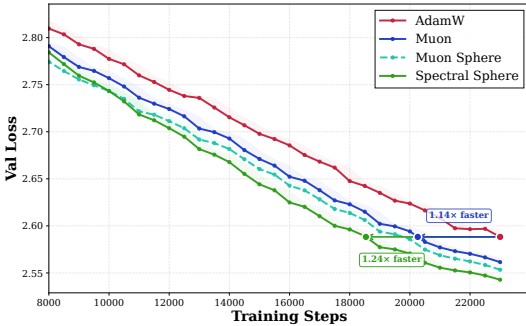

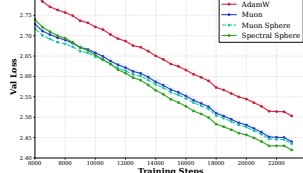

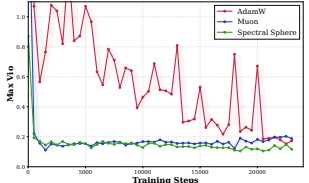

*(a)* Validation Loss across four optimizers.

*(b)* MaxVio as a MoE load-balance metric.

*Figure 8.* **MoE 8B-A1B training.** Spectral Sphere achieves the lowest validation loss while maintaining the best load balancing. In contrast, AdamW exhibits substantially larger MaxVio with frequent spikes, indicating unstable routing and poorer utilization of effective model capacity. Compared to Muon, constraining each expert on the spectral sphere further improves load balance.

*Figure 7.* **Validation loss of training dense 1.7B model on 100B tokens.** As a reference point, AdamW attains a final validation loss of 2.588 at 23k steps. The overall setup favors AdamW, since the learning rate is tuned for it (5e-3), rather than the higher optimal rate (1e-2) used for Muon and Spectral Sphere. Even under this setting, spectral-based optimizers exhibit higher efficiency: Muon reaches the same validation loss level in 12% fewer steps, while Spectral Sphere does so in 19% fewer steps.

pert load balancing compared to global auxiliary loss (Qiu et al., 2025) in our ablations. We enable expert bias with an update rate of 0.001. The sequence-level auxiliary loss is removed, as we find it redundant when expert bias is enabled. We use a sigmoid gate with top-$k$ scores renormalized and scaled by 2. (see Appendix A.4). To evaluate router load balancing, we employ Maximal Violation (MaxVio) (Wang et al., 2024), where 0 indicates perfect balance. As shown in Figure 8, weight spectral normalization effectively promotes balanced routing, leading to superior validation loss.

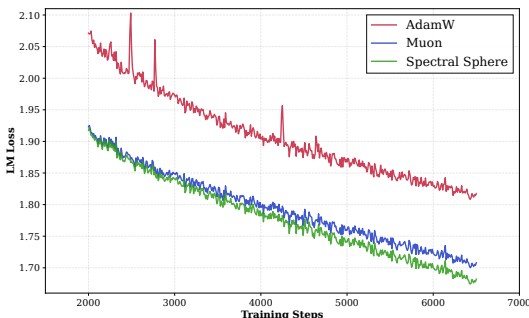

*Figure 9.* **Deepnet 200 layers training loss.** AdamW shows occasional instability, characterized by several loss spikes and a growing gap in performance relative to spectral-based optimizers. Spectral Sphere attains both the lowest loss and highest stability.

### 6.4. DeepNet 200-Layer

To evaluate the stability of different optimizers under extreme depth, we extended the baseline's 28 layers to 200 layers. This serves as a stress test for stability. The training loss in Figure 9 shows that Spectral Sphere outperforms baselines with lower loss and higher stability.

## 7. Discussion

In this work, we propose a novel perspective on optimizer design grounded in spectral $\mu$P principles. By identifying the spectral sphere as the natural geometry for sta-

ble feature learning, we derive the Spectral Sphere Optimizer (SSO) — the unique solution for steepest descent constrained within both weight and update manifolds (Section 3). This formulation effectively achieves rapid convergence grounded in fundamental training stability. Empirically, SSO consistently outperforms AdamW and Muon, while uniquely preserving stable $\mu$P learning rate transfer.

A critical distinction exists between our approach and emerging works on manifold optimization. While Stiefel manifold (Bernstein, 2025) requires *all* singular values to be exactly 1 (overly rigid), and Hyperball (Wen et al., 2025) constrains the Frobenius norm (overly loose, permitting a single singular value to grow disproportionately large),

SSO constrains only the *maximal* singular value. This strikes an ideal balance: it provides the strict worst-case activation bounds required by $\mu$P while allowing the internal spectrum to evolve freely below the spectral norm bound.

Beyond the theoretical contribution, we provide a complete and systematic recipe (Section 4) implemented in Megatron-LM, which serves as a robust empirical practice for the broader class of spectral optimizers. While the current root solver introduces non-trivial latency, we outline concrete pathways to mitigate this in Appendix A.6. For scenarios prioritizing infra cost, we recommend Muon-Sphere — a variant that retains equivalent activation control with minimal overhead.

## Acknowledgements

We sincerely thank Jianlin Su for his excellent blog, Scientific Spaces. We also thank Yifei Shen for his thoughtful discussions, and Franz Louis Cesista for helping clarify several conceptual formulations.

## Impact Statement

This paper presents work whose goal is to advance the field of Machine Learning. There are many potential societal consequences of our work, none which we feel must be specifically highlighted here.

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

# A. Appendix

## A.1. Duality with Spectral Norm

We may first prove Theorem A.1, and we can then directly derive Equation (11). And Equation (12) comes from the constraint in Equation (10).

**Theorem A.1.** *Nuclear norm* $\|\cdot\|_*$ *is the dual norm of spectral norm* $\|\cdot\|_2$, *which means*

$$\|\boldsymbol{G}\|_* = \max_{\|T\|_2=1} \langle \boldsymbol{G}, \boldsymbol{T} \rangle.$$

$\mathrm{msign}(\cdot)$ *is the dual map based on* $\|\cdot\|_2$, *which means*

$$\mathrm{msign}(\boldsymbol{G}) = \operatorname*{argmax}_{\|\boldsymbol{T}\|_2=1} \langle \boldsymbol{G}, \boldsymbol{T} \rangle.$$

*Proof.* Since $\boldsymbol{G} \in \mathbb{R}^{n \times m}$ has Singular Value Decomposition $\boldsymbol{G} = \boldsymbol{U}\boldsymbol{\Sigma}\boldsymbol{V}^\top = \sum_{i=1}^r \sigma_i \boldsymbol{u}_i \boldsymbol{v}_i^\top$ where $\boldsymbol{u}_i \in \mathbb{R}^n$ and $\boldsymbol{v}_i \in \mathbb{R}^m$ are left and right singular vectors, we have

$$\langle \boldsymbol{G}, \boldsymbol{T} \rangle = \mathrm{Tr}(\boldsymbol{G}^\top \boldsymbol{T}) = \mathrm{Tr}(\sum_{i=1}^r \sigma_i \boldsymbol{v}_i \boldsymbol{u}_i^\top \boldsymbol{T}) = \sum_{i=1}^r \sigma_i \boldsymbol{u}_i^\top \boldsymbol{T} \boldsymbol{v}_i. \tag{17}$$

With $\|\boldsymbol{T}\|_2 = 1$ and $\|\boldsymbol{u}_i\|_2 = \|\boldsymbol{v}_i\|_2 = 1$, we have $\boldsymbol{u}_i^\top \boldsymbol{T} \boldsymbol{v}_i \leq \|\boldsymbol{T}\|_2 \|\boldsymbol{u}_i\|_2 \|\boldsymbol{v}_i\|_2 = 1$, hence

$$\langle \boldsymbol{G}, \boldsymbol{T} \rangle = \sum_{i=1}^r \sigma_i \boldsymbol{u}_i^\top \boldsymbol{T} \boldsymbol{v}_i \leq \sum_{i=1}^r \sigma_i = \|\boldsymbol{G}\|_*. \tag{18}$$

Equality is attained when $\boldsymbol{u}_i^\top \boldsymbol{T} \boldsymbol{v}_i = 1$ for all $i = 1, \dots, r$, that is when

$$\boldsymbol{T} = \sum_{i=1}^r \boldsymbol{u}_i \boldsymbol{v}_i^\top = \boldsymbol{U}_{[:,:r]} \boldsymbol{V}_{[:,:r]}^\top = \mathrm{msign}(\boldsymbol{G}). \tag{19}$$

Note that for this $\boldsymbol{T}$ we indeed have $\|\boldsymbol{T}\|_2 = 1$ (its nonzero singular values are all equal to 1), so $\boldsymbol{T}$ is feasible and achieves the upper bound. Therefore, we have

$$\operatorname*{argmax}_{\|\boldsymbol{T}\|_2=1} \langle \boldsymbol{G}, \boldsymbol{T} \rangle = \mathrm{msign}(\boldsymbol{G}), \tag{20}$$

and according to Equation (18), the maximum value equals $\|\boldsymbol{G}\|_*$. $\qquad\square$

## A.2. Proofs: Localization of the Root of $h(\lambda)$

In this section, we first prove the localization of the root $\lambda^\star$ to $h(\lambda) = 0$, which is required in Section 3.2. We then present experiments showing that the computational overhead of the proposed $\lambda$-solver is negligible compared to the whole training process.

We first prove Theorem A.2, from which Theorem A.3 follows.

**Theorem A.2.** *The function*

$$h(\lambda) = \langle \boldsymbol{\Theta}, \boldsymbol{\Phi}^\star(\lambda) \rangle = \langle \boldsymbol{\Theta}, \mathrm{msign}(\boldsymbol{G} + \lambda \boldsymbol{\Theta}) \rangle$$

*is monotonic non-decreasing in* $\lambda$.

*Proof.* Recall that the matrix sign function can be equivalently defined as the solution of the spectral-norm constrained maximization as in Theorem A.1:

$$\mathrm{msign}(\boldsymbol{G}) = \operatorname*{argmax}_{\|T\|_2=1} \langle \boldsymbol{G}, \boldsymbol{T} \rangle. \tag{21}$$

Consider two values $\lambda_2 > \lambda_1$. By the definition of $\boldsymbol{\Phi}^\star(\lambda)$ as the maximizer of $\langle \boldsymbol{G} + \lambda\boldsymbol{\Theta}, \cdot \rangle$, we obtain

$$\langle \boldsymbol{G} + \lambda_1\boldsymbol{\Theta}, \boldsymbol{\Phi}^\star(\lambda_1)\rangle \geq \langle \boldsymbol{G} + \lambda_1\boldsymbol{\Theta}, \boldsymbol{\Phi}^\star(\lambda_2)\rangle$$
$$\Rightarrow \quad \langle \boldsymbol{G}, \boldsymbol{\Phi}^\star(\lambda_1)\rangle + \lambda_1\langle\boldsymbol{\Theta}, \boldsymbol{\Phi}^\star(\lambda_1)\rangle \geq \langle \boldsymbol{G}, \boldsymbol{\Phi}^\star(\lambda_2)\rangle + \lambda_1\langle\boldsymbol{\Theta}, \boldsymbol{\Phi}^\star(\lambda_2)\rangle, \tag{22}$$
$$\langle \boldsymbol{G} + \lambda_2\boldsymbol{\Theta}, \boldsymbol{\Phi}^\star(\lambda_2)\rangle \geq \langle \boldsymbol{G} + \lambda_2\boldsymbol{\Theta}, \boldsymbol{\Phi}^\star(\lambda_1)\rangle$$
$$\Rightarrow \quad \langle \boldsymbol{G}, \boldsymbol{\Phi}^\star(\lambda_2)\rangle + \lambda_2\langle\boldsymbol{\Theta}, \boldsymbol{\Phi}^\star(\lambda_2)\rangle \geq \langle \boldsymbol{G}, \boldsymbol{\Phi}^\star(\lambda_1)\rangle + \lambda_2\langle\boldsymbol{\Theta}, \boldsymbol{\Phi}^\star(\lambda_1)\rangle. \tag{23}$$

Summing Equation (22) and Equation (23), we can derive

$$\lambda_1\langle\boldsymbol{\Theta}, \boldsymbol{\Phi}^\star(\lambda_1)\rangle + \lambda_2\langle\boldsymbol{\Theta}, \boldsymbol{\Phi}^\star(\lambda_2)\rangle \geq \lambda_1\langle\boldsymbol{\Theta}, \boldsymbol{\Phi}^\star(\lambda_2)\rangle + \lambda_2\langle\boldsymbol{\Theta}, \boldsymbol{\Phi}^\star(\lambda_1)\rangle \tag{24}$$
$$\Rightarrow \quad (\lambda_2 - \lambda_1)\langle\boldsymbol{\Theta}, \boldsymbol{\Phi}^\star(\lambda_2)\rangle \geq (\lambda_2 - \lambda_1)\langle\boldsymbol{\Theta}, \boldsymbol{\Phi}^\star(\lambda_1)\rangle \quad \Rightarrow \quad \langle\boldsymbol{\Theta}, \boldsymbol{\Phi}^\star(\lambda_2)\rangle \geq \langle\boldsymbol{\Theta}, \boldsymbol{\Phi}^\star(\lambda_1)\rangle, \tag{25}$$

with $\lambda_2 - \lambda_1 > 0$. Hence, $h(\lambda) = \langle\boldsymbol{\Theta}, \mathrm{msign}(\boldsymbol{G} + \lambda\boldsymbol{\Theta})\rangle$ is monotonic non-decreasing in $\lambda$. $\qquad\square$

**Theorem A.3.** *There exists some $\lambda^\star \in \mathbb{R}$ such that $h(\lambda^\star) = 0$, and any such root satisfies*

$$|\lambda^\star| \leq 2\|\boldsymbol{G}\|_*.$$

*Proof.* We first prove the existence of a root $\lambda^\star$. Since $\boldsymbol{\Theta} = \boldsymbol{u}_1\boldsymbol{v}_1^\top$ is a rank-one matrix with unit-norm singular vectors, its nuclear norm satisfies $\|\boldsymbol{\Theta}\|_* = 1$. Moreover, by construction, we have

$$\|\boldsymbol{\Phi}^\star(\lambda)\|_2 = \|\mathrm{msign}(\boldsymbol{G} + \lambda\boldsymbol{\Theta})\|_2 = 1. \tag{26}$$

By Theorem A.1, letting $\boldsymbol{T}_1 = \frac{\boldsymbol{\Phi}^\star(\lambda)}{\|\boldsymbol{\Phi}^\star(\lambda)\|_2}, \boldsymbol{T}_2 = -\frac{\boldsymbol{\Phi}^\star(\lambda)}{\|\boldsymbol{\Phi}^\star(\lambda)\|_2}$, we have

$$\|\boldsymbol{T}_1\|_2 = \|\boldsymbol{T}_2\|_2 = 1 \quad \text{and thus} \quad \langle\boldsymbol{\Theta}, \boldsymbol{T}_1\rangle \leq \|\boldsymbol{\Theta}\|_* \text{ and } \langle\boldsymbol{\Theta}, \boldsymbol{T}_2\rangle \leq \|\boldsymbol{\Theta}\|_*. \tag{27}$$

Therefore, we have

$$|\langle\boldsymbol{\Theta}, \mathrm{msign}(\boldsymbol{G} + \lambda\boldsymbol{\Theta})\rangle| \leq \|\boldsymbol{\Theta}\|_* \|\mathrm{msign}(\boldsymbol{G} + \lambda\boldsymbol{\Theta})\|_2 = 1, \tag{28}$$

which implies $|h(\lambda)| \leq 1$ for all $\lambda \in \mathbb{R}$. Moreover, since $h(\lambda)$ is monotonic non-decreasing by Theorem A.2, the limits $\lim_{\lambda\to-\infty} h(\lambda)$ and $\lim_{\lambda\to+\infty} h(\lambda)$ exist. Using the property $\mathrm{msign}(\lambda\boldsymbol{T}) = \mathrm{msign}(\boldsymbol{T})$, we have

$$\lim_{\lambda\to+\infty} \mathrm{msign}(\boldsymbol{G} + \lambda\boldsymbol{\Theta}) = \lim_{\lambda\to+\infty} \mathrm{msign}\left[\lambda\left(\boldsymbol{\Theta} + \tfrac{1}{\lambda}\boldsymbol{G}\right)\right] = \mathrm{msign}(\boldsymbol{\Theta}) = \boldsymbol{\Theta}, \tag{29}$$

where the last equality is satisfied by definition of $\mathrm{msign}(\cdot)$ and $\boldsymbol{\Theta}$. Consequently,

$$\lim_{\lambda\to+\infty} h(\lambda) = \langle\boldsymbol{\Theta}, \boldsymbol{\Theta}\rangle = \mathrm{Tr}(\boldsymbol{v}_1\boldsymbol{u}_1^\top\boldsymbol{u}_1\boldsymbol{v}_1^\top) = 1. \tag{30}$$

Similarly, we can also obtain $\lim_{\lambda\to-\infty} h(\lambda) = -1$. Since $\mathrm{msign}(\boldsymbol{G} + \lambda\boldsymbol{\Theta})$ is a subgradient of a convex function $\|\boldsymbol{G} + \lambda\boldsymbol{\Theta}\|_*$ with respect to $\lambda$ according to Watson (1992), monotonic non-decreasing $h(\lambda)$ satisfies the intermediate value property. Consequently, there exists at least one $\lambda^\star \in \mathbb{R}$ such that $h(\lambda^\star) = 0$.

We next localize the root. For any $\lambda > 2\|\boldsymbol{G}\|_* > 0$ and any matrix $\boldsymbol{T}$ with $\|\boldsymbol{T}\|_2 = 1$, we have

$$\lambda\langle\boldsymbol{\Theta}, \boldsymbol{T}\rangle = \langle\boldsymbol{G} + \lambda\boldsymbol{\Theta}, \boldsymbol{T}\rangle - \langle\boldsymbol{G}, \boldsymbol{T}\rangle. \tag{31}$$

By Equation (27), $\langle\boldsymbol{G}, \boldsymbol{T}\rangle \leq \|\boldsymbol{G}\|_*$. Let $\boldsymbol{T} = \mathrm{msign}(\boldsymbol{G} + \lambda\boldsymbol{\Theta})$, using Theorem A.1, we can derive

$$\lambda h(\lambda) = \|\boldsymbol{G} + \lambda\boldsymbol{\Theta}\|_* - \langle\boldsymbol{G}, \boldsymbol{T}\rangle \geq \left|\|\boldsymbol{G}\|_* - \lambda\|\boldsymbol{\Theta}\|_*\right| - \|\boldsymbol{G}\|_* \geq \lambda - 2\|\boldsymbol{G}\|_* > 0, \tag{32}$$

and this implies $h(\lambda) > 0$. Similarly, if $\lambda < -2\|\boldsymbol{G}\|_*$, $h(\lambda) < 0$. Since $h(\lambda)$ is monotonic non-decreasing, any root $\lambda^\star$ satisfying $h(\lambda^\star) = 0$ must therefore lie in the interval $[-2\|\boldsymbol{G}\|_*, 2\|\boldsymbol{G}\|_*]$. $\qquad\square$

Theorem A.3 actually provides a theoretical guarantee that the correctness of using the bisection algorithm to solve the root $\lambda^\star$. To further validate this theoretical result, we randomly generate several zero-centered matrices with varying dimensions, construct $\boldsymbol{G}$ and $\boldsymbol{\Theta}$ as described in this section, and plot the curve of $h(\lambda)$ together with its root. As shown in Figure 3, $h(\lambda)$ is indeed monotonic in $\lambda$, and its root lies close to $\lambda = 0$, which is consistent with our theoretical analysis.

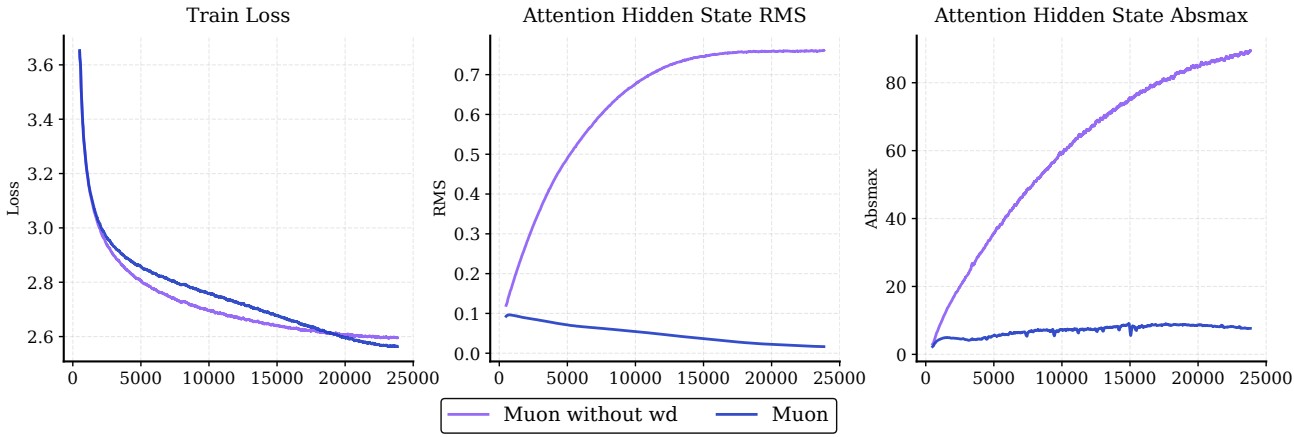

*Figure 10.* **Ablation of weight decay in Muon.** In contrast with Spectral Sphere (Figure 1), without weight constraint, Muon training dynamics become unstable, which in-turn would hurt performance.

### A.3. Dynamic Spectral Weight Decay

In this section, we introduce spectral retraction mechanisms to counteract the accumulation of higher-order errors that may gradually drift the weights off the spectral sphere. As noted in Section 3.3, the remainder term $\mathcal{O}(\eta^2 R^2 \|\Phi\|_2^2)$ in the Taylor expansion can accumulate over iterations. To enforce the exact constraint $\|W\|_2 = R$ throughout training, we apply Equation (13) that projects the weights back onto the spectral manifold.

In practice, we consider two retraction variants that differ in how strictly the spectral constraint is enforced. The hard variant applies an explicit projection to maintain $\|W\|_2 = R$ at every step, while the dynamic variant replaces the exact projection with a soft, learning-rate-scaled radial correction that steers $\|W\|_2$ toward the target radius. These two variants correspond to enforcing the spectral constraint exactly or approximately, and lead to different interactions with conventional weight decay.

**Hard retraction (exact projection).** The hard variant applies the retraction map explicitly using an estimated top singular value $\sigma \approx \|W\|_2$ which is computed via Power Iteration.

$$W \leftarrow \frac{R}{\sigma} W. \tag{33}$$

This enforces the exact constraint $\|W\|_2 = R$ at every step, with deviations arising solely from the approximation error in $\sigma$. Under this setting, Spectral Sphere is typically used together with standard decoupled weight decay as in AdamW: weight decay shrinks the full parameter vector, while the spectral retraction constrains the dominant singular value.

**Dynamic retraction (soft spectral decay).** The dynamic variant replaces the exact projection with a small, sign-controlled radial adjustment with hyperparameter $\lambda$.

$$W \leftarrow \big(1 + \lambda \eta \operatorname{sign}(R - \sigma)\big) W. \tag{34}$$

Instead of strictly enforcing the constraint $\|W\|_2 = R$, this update gently adjusts the spectral norm toward the target radius $R$ based on $\sigma$. Since the correction is proportional to $\eta$, the adjustment strength diminishes automatically as learning rate schedules. Under this formulation, dynamic retraction functions as a spectrally aligned, layer-wise analogue to decoupled weight decay. Consequently, we tie $\lambda$ to the AdamW weight decay coefficient and employ the dynamic variant as a drop-in replacement for conventional weight decay.

From a geometric perspective, the two variants above can both be viewed as approximate projections onto the spectral sphere. As discussed in Section 3.3, applying retraction at the beginning of iteration $t + 1$ instead of after the update at iteration $t$ only introduces $\mathcal{O}(\eta^2)$ discrepancies. Consequently, the update remains first-order equivalent to steepest descent on the spectral manifold.

## A.4. MoE Routing Scaling Factor

Following (Su, 2025c), in our MoE architecture, the output $\boldsymbol{y}$ is the sum of the shared experts and the routed experts. A critical issue arises from the magnitude mismatch between the two parts:

$$\boldsymbol{y} = \underbrace{\sum_{i=1}^{N_{\text{shared}}} \boldsymbol{e}_{\text{shared},i}}_{\text{Weight} \approx 1} + \underbrace{\sum_{j \in \text{TopK}} g_j \boldsymbol{e}_{\text{routed},j}}_{\text{Weight } g_j \ll 1} \tag{35}$$

The shared experts are directly added to the residual stream, effectively having a coefficient of 1. In contrast, the routed experts are multiplied by sigmoid probabilities $g_j$, which are typically small. Consequently, the variance (signal energy) of the routed experts is significantly lower than that of the shared experts, causing the optimizer to neglect the routed part.

To balance the contributions, we introduce a scaling factor $M$ to the routed branch (Su, 2025c):

$$\boldsymbol{y} = \sum_{i=1}^{N_{\text{shared}}} \boldsymbol{e}_{\text{shared},i} + M \sum_{j \in \text{TopK}} g_j \boldsymbol{e}_{\text{routed},j} \tag{36}$$

We aim to choose $M$ such that the expected variance of the routed term matches that of the shared term. Assuming experts have unit variance:

$$M \approx \sqrt{\frac{N_{\text{shared}}}{\mathbb{E}[\sum g_j^2]}} \tag{37}$$

Using numerical simulation (Listing 1) with $N_{\text{shared}} = 1$ and Top-4 sigmoid routing, we find $M \approx 2.0$. We find this scaling factor is crucial for MoE training stability.

*Listing 1.* Python implementation for estimating the MoE scaling factor $M$.

```python
import numpy as np

def estimate_scaling_factor(n_total=64, k_routed=4, n_shared=1):
    factors = []
    for _ in range(10000):
        # 1. Simulate logits
        logits = np.random.randn(n_total - n_shared)

        # 2. Get Sigmoid Scores
        scores = 1 / (1 + np.exp(-logits))

        # 3. Select Top-K routed scores
        scores = np.sort(scores)[::-1][:k_routed]

        # 3. Normalize scores
        scores /= scores.sum()

        # 4. Calculate Magnitude
        magnitude = np.sum(scores**2)**0.5

        factors.append(n_shared**0.5 / magnitude)

    return np.mean(factors)
```

## A.5. $\mu$P Width Scaling

In Figure 2, we conduct experiment to validate $\mu$P width scaling across different model sizes from 70M to 1.8B. Additional AdamW results are included in Figure 11. We scale the hidden size, intermediate size, and number of attention heads (while fixing head dimensions). The models are trained on 30B tokens sampled from the Olmo2-mix-1124 dataset (Team OLMo et al., 2025). All optimizers use the Spectral $\mu$P LR Scaler.

- The LR is swept across {1e-3, 3e-3, 5e-3, 7e-3, 9e-3, 1e-2, 1.5e-2, 2e-2, 3e-2}.

- Hidden size is swept across {256, 512, 1024, 2048}.

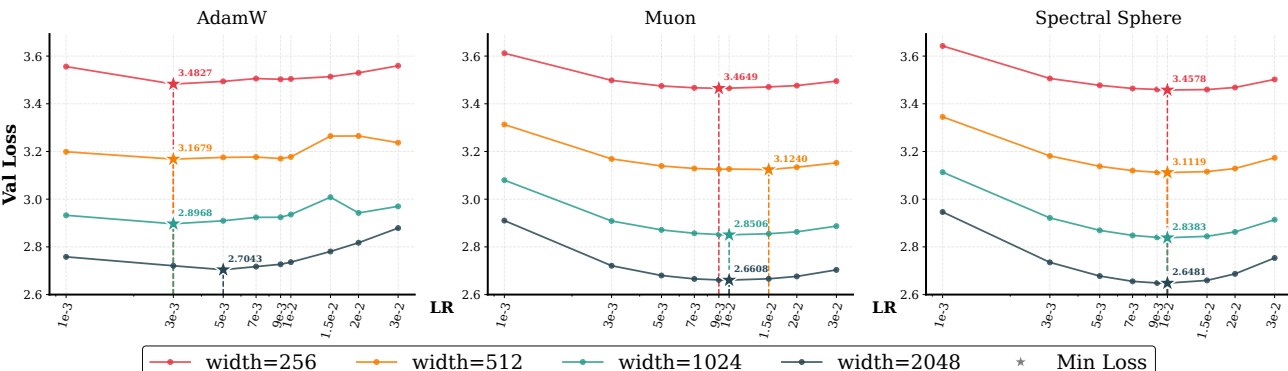

*Figure 11.* **$\mu$P LR grid search with AdamW, Muon and Spectral Sphere.** AdamW shows even worse validation loss, with drifting optimal LR.

## A.6. Future Improvements

In Section 5, we introduce the infrastructure design of SSO. Several potential improvements are discussed here.

- **GPU-Native Solver.** Profiling indicates that the current CPU-based bisection solver introduces latency due to frequent device-host synchronization. Although the bracketing phase converges rapidly ($< 2$ steps), the bisection phase averages 5–7 steps, creating synchronization bubbles. Future work will prioritize implementing a pure GPU-native solver to eliminate these overheads. Additionally, we plan to adopt higher convergence algorithms, such as Brent's method or $n$-section search, and optimize initial bracketing intervals to minimize msign calls.

- **Kernel Optimization.** The theoretical advantage of SSO relies on the accuracy of the tangent space projection; when errors are high, the update direction may degrade to the "worst-case" Muon update. Currently, we ensure precision via 8 iterations of msign in FP32. To follow standard Muon practices, we plan to use msign in BFloat16 within 5 steps. Furthermore, we intend to replace current JIT-compiled operations with custom, fully optimized kernels (e.g. for batched msign and Power Iterations) to better exploit the hardware features of next-generation GPUs.

- **Low-precision Training.** While weight matrices are spectral constrained, we find the residual stream remains the primary source of outliers. We aim to explore "fully manifold constrained" architectures, i.e. using mHC (Xie et al., 2025). Additionally, given SSO's demonstrated stability, we plan to explore low-precision training (e.g. FP8/NVFP4) to leverage the high throughput of low-bit arithmetic for better training efficiency.

