# OpenReview forum: "Controlled LLM Training on Spectral Sphere"
_ICML.cc/2026/Conference — ICML 2026 spotlight_

### Official Review · Reviewer_CZto · 2026-03-12

**Soundness:** 3
**Presentation:** 3
**Significance:** 4
**Originality:** 3
**Overall Recommendation:** 5
**Confidence:** 4

**Summary:**

This paper proposes a new optimizer, the Spectral Sphere Optimizer (SSO), which provably satisfies the steepest-descent property and the muP spectral condition. The paper also empirically shows that SSO consistently outperforms AdamW and Muon in a series of pretraining experiments at scale.

**Compliance With Llm Reviewing Policy:**

Affirmed.

**Ethical Review Concerns:**

No ethical review needed

**Final Justification:**

As I noted in my initial review, the greatest strength of this paper is the strong alignment between theory and practice. As an optimizer derived from muP theory, SSO is empirically shown to exhibit good hyperparameter transferability. In addition, SSO removes the weight-decay coefficient, which further simplifies hyperparameter tuning. Therefore, I am willing to maintain my original rating.

**Key Questions For Authors:**

Some recent work has explored norm constraints other than the spectral norm. For example, a recent approach proposes a new optimizer called Hyperball based on an F-norm constraint [1]. I am curious whether different norm constraints lead to meaningful differences in practice and in the corresponding theoretical analysis. (btw I am just curious and this question won't affect my rating.)

[1] https://psychedelic-sunstone-851.notion.site/Fantastic-Pretraining-Optimizers-and-Where-to-Find-Them-2-1-Hyperball-Optimization-2e924306e6f280e7a5ffee00eb40a0dd(https://psychedelic-sunstone-851.notion.site/Fantastic-Pretraining-Optimizers-and-Where-to-Find-Them-2-1-Hyperball-Optimization-2e924306e6f280e7a5ffee00eb40a0dd)

**Limitations:**

yes

**Strengths And Weaknesses:**

### Strengths：
* The derivation of SSO is clear and theoretically grounded.
* The greatest strength of this paper is the strong alignment between theory and practice. As an optimizer derived from muP theory, SSO is empirically shown to exhibit good hyperparameter transferability. In addition, SSO removes the weight-decay coefficient, which further simplifies hyperparameter tuning.
* SSO is shown to outperform AdamW and Muon—the two most prominent optimizers in current pretraining research—in at-scale experiments, including a 1.7B dense model and an 8B MoE model. These results provide convincing evidence of its effectiveness.
### Weaknesses:
* Because transformer-based architectures are not fully scale-invariant, I am not sure whether fully constraining the weight norm could limit model capacity. For example, attention mechanisms often require relatively large logits.

---

> ### Author Rebuttal · Authors · 2026-03-30
>
> We would like to thank the reviewer for the positive feedback. We address your insightful comments below:
>
> **Response to Weakness: Does fully constraining the weight norm limit model capacity?**
> Empirically, we observe no capacity degradation with SSO; in fact, it consistently matches or exceeds the baselines across all downstream benchmarks (Table 3) and achieves lower validation loss.
>
> Theoretically, we argue that constraining weight spectral norms does not fundamentally limit the representational capacity of hidden states. A hidden state vector can encode all necessary information in its direction alone — the norm is not an indispensable degree of freedom. Even when scalar magnitude information is needed, the model can dedicate a subset of hidden dimensions to encode it explicitly, much like positional encodings use vector to encode a continuous scalar. Therefore, fixing the spectral scale of weight matrices does not fundamentally reduce the information capacity of the network; it merely removes a redundant, unconstrained degree of freedom that is otherwise a source of training instability.
>
> However, your intuition regarding operations that are not fully scale-invariant—such as the attention mechanism—is very sharp. The only place where constraining the norm might theoretically restrict expressivity is indeed right before the softmax operation, where the scale of the logits directly acts as an inverse temperature, controlling the sharpness of the attention distribution. Currently, the $\mu$P framework itself primarily focuses on linear projections and does not provide rigorous theoretical guarantees for non-linearities or deep compositional dynamics.
>
> To address the specific issue of attention logits in the future, a promising improvement would be to introduce a learnable scalar (a temperature parameter) immediately before the softmax. This would decouple the sharpness of the attention distribution from the constrained weight norm, allowing the model to learn sharp attention patterns without violating the spectral constraints on the weight matrices themselves.
>
> From a broader perspective on expressivity and scaling, we hold the position that rather than scaling models in an unconstrained setting that is difficult to mathematically characterize, it is far more beneficial to perform *controlled scaling* under strict theoretical constraints. By doing so, we avoid entangling the training dynamics with complex, coupled hyperparameters. We believe this constrained approach yields a cleaner, more interpretable framework that is better suited for theoretical analysis and can more reliably guide the next generation of scaling laws.
>
> **Q: Spectral norm vs. Frobenius norm (e.g., Hyperball)**
> This is an excellent question. The choice of the spectral norm in SSO is deeply rooted in the $\mu$P theory. The spectral norm ($\|W\|_2$) corresponds exactly to the $L_2 \to L_2$ (or RMS-to-RMS) operator norm, which provides the tightest and most direct bound on the worst-case activation gain across a layer (as discussed in Section 2.1).
>
> In contrast, a Frobenius norm constraint (like in Hyperball) is mathematically looser. Since the Frobenius norm is the $L_2$ norm of the vector of singular values ($\|W\|_F = \sqrt{\sum \sigma_i^2}$), it only bounds the *sum of squares* of the singular values. Under a Frobenius constraint, it is entirely possible for a single top singular value ($\sigma_1$) to grow disproportionately large while the others shrink, as long as the total sum remains bounded. This means a Frobenius constraint cannot strictly prevent the emergence of dominant singular directions, which are often the culprit behind activation spikes and training instabilities in LLMs.
>
> Therefore, while the Frobenius norm is easier to compute, the spectral norm provides the strict worst-case guarantees required by $\mu$P. A systematic empirical and theoretical comparison of these different norm constraints (and their impact on the eigenspectrum dynamics during training) is indeed a fascinating direction for future work. We will add a brief discussion of this comparison in the revision.

---

> > ### Author Rebuttal · Reviewer_CZto · 2026-04-03
> >
> > Thanks for the detailed response. I will keep my score.

---

### Official Review · Reviewer_xbZJ · 2026-03-12

**Soundness:** 3
**Presentation:** 3
**Significance:** 3
**Originality:** 4
**Overall Recommendation:** 5
**Confidence:** 4

**Summary:**

When training machine learning models, there is a constant interplay between quality and scalability. For the former, we wish to produce better quality optimizers, most recently, Muon, that allow models to reach better solutions in fewer iterations. For the latter, we want the insights and hyperparameters found on small-scale problems to scale robustly across model sizes, which led to the invention of frameworks such as $\mu$P.

However, Muon is only “half-aligned” to $\mu$P conditions: while update scales are controlled with increasing model size, the norm of the weights is not. This unconstrained weight drift leads to activation instability. To address this, the paper proposes the Spectral Sphere Optimizer (SSO). SSO formulates optimization as a steepest descent problem constrained to a module-wise "spectral sphere," where both weights and updates are strictly bound by a target spectral radius. The method utilizes a Lagrange multiplier search via a bracket-and-bisect solver to find the update direction in the tangent space, followed by a retraction step to maintain the weight constraint. To make this feasible at scale, the authors implement SSO using techniques like atomic module sharding and adaptive kernel selection to minimize the solver overhead. Evaluation across a dense and MoE architectures shows that SSO not only outperforms AdamW and Muon but also yields practical stability benefits.

**Compliance With Llm Reviewing Policy:**

Affirmed.

**Final Justification:**

My final recommendation is to advocate for the acceptance of this work. As stated in my original review: "this paper is a very strong contribution both theoretically and empirically and would offer deep insights that are of interest to the machine learning community.". In addition, the rebuttal addressed the additional clarifying points that had raised to a high level. As such, this reinforced my original position and I wish to maintain my initial score.

**Key Questions For Authors:**

1. While Section 3.3 introduces the "MuonSphere" baseline to empirically show that heuristic projections perform worse than SSO, could you provide further theoretical justification for *why* these heuristic methods fail to align with the desired training dynamics compared to your strict steepest descent on the spectral sphere?
2. Can you provide an ablation of the retraction step for consistency so that you have empirical proofs to assist with the mathematical derivation? Even if these experiments are on a small scale, it would be useful to show that without this property, the method would fail to align with the $\mu$P framework.
3. Acknowledging your note on the current latency overheads, could you please provide an analysis of your method against Muon and AdamW on a wall-clock time axis? This would greatly improve the practical context of the paper beyond the step-wise performance benefit.
4. Hyperparameter guidance: Does an optimal $c$ transfer across architecture sizes, and is there any theoretical heuristic you can suggest to determine an optimal value other than searching across a set of candidate values? Furthermore, in the paper, the authors suggest that the module granularity should be set based on infrastructure speed. Can the authors offer any heuristics of how to do so, perhaps motivated by a theoretical modelling of the infrastructure conditions?
5. In Figure 11, the AdamW baseline exhibits significant width-dependent variance in activation scales. Is this baseline implemented using Standard Parametrization (SP) or Maximal Update Parametrization ($\mu$P)? If it is $\mu$P, why do we see a lack of width-invariance (overlap), and does this suggest that weight drift in AdamW is sufficient to invalidate $\mu$P scaling even early in training?

**Limitations:**

No, the authors have not done so explicitly. However, given the quality of the paper, there is nothing specific that I feel necessary to raise other than the comments I have raised in the preceding sections.

**Strengths And Weaknesses:**

**Overall Assessment:**

This paper is a very strong contribution both theoretically and empirically and would offer deep insights that are of interest to the machine learning community. The paper would benefit from extra clarity on certain design decisions and extra ablations to complete the full understanding. However, these would only improve an already well-executed piece of research.

**Presentation**

- **Strength:** The paper is well presented. It is well contextualized in the modern machine learning landscape. The delivery of SSO is well-structured and clearly motivated, supported by both mathematical and empirical evidence.
- **Weakness:** Nit: please could the authors consider improving the legibility of their figures. While the content is great, some of the figures and their axes labels are small. Examples are Figure 1 and Figure 3; this would improve the visual aspect and readability of the paper significantly.

**Soundness**

- **Strengths:** The submission is very technically sound. The improvement over Muon is clear; it is theoretically motivated and empirically validated. The baselines chosen and experimental framework are convincing to motivate the benefit of using their method.
- **Nit:** For consistency purposes, the paper would benefit from empirical evidence or a citation that motivates the assumption that the top singular vectors move slowly throughout training.

**Significance**

- **Strength:** The research is significant given that it addresses a key issue in scaling Muon and further improving its empirical performance. Furthermore, the optimizer demonstrates improvements for both dense and MoE architectures which would be of great interest to the machine learning community.
- **Weakness:** Although the paper discusses the latency impact for SSO in Tables 1 and 2 and transparently acknowledges the current overhead of synchronization bubbles, the performance is currently primarily shown as a function of step count. It would be more convincing to show the performance of the optimizer on a wall-clock time axis. This comparison would improve the significance as it creates a greater understanding for practitioners regarding the this Pareto tradeoff.

**Originality**

• **Strength:** The mathematical investigation into the link between Muon and $\mu$P is novel and well-executed.

• **Weakness:** n/a

---

> ### Author Rebuttal · Authors · 2026-03-30
>
> We would like to thank the reviewer for the careful evaluation and supportive comments! We respectfully provide our response as follows.
>
> **Q1: Why do heuristic projections (MuonSphere) underperform SSO?**
>
> While both MuonSphere and SSO constrain their updates to the tangent space (satisfying the constraint in Eq. 10), SSO analytically solves for the *exact steepest descent direction* that maximizes loss reduction within this constrained space. MuonSphere, on the other hand, is a suboptimal heuristic. Intuitively, MuonSphere takes a standard unconstrained step and then projects back to the sphere, which artificially shrinks the effective step size and deviates from the true steepest descent path on the manifold. SSO walks optimally along the spherical constraint from the very start, yielding the consistent gains observed in our experiments.
>
>  **Q2: Consistency of the Retraction Step**
>
> Thank you for pointing this out! Our method explicitly applies the spectral norm retraction at every single step, constantly re-scaling the top singular value to the target radius $R = c\sqrt{d_{out}/d_{in}}$. This ensures the top singular vectors move slowly throughout training. Empirically, we conservatively set the number of power iterations to 20, which is sufficient to ensure numerical convergence of the spectral norm.
>
> **Q3: Wall-clock time analysis vs. AdamW and Muon**
>
> We completely agree that wall-clock time is a crucial metric for practitioners. Currently, in our 1.7B setup, SSO converges in 19% fewer steps than AdamW and 9% fewer steps than Muon. However, Muon converges slightly faster in pure wall-clock time due to SSO's ~11.45% per-step overhead (Table 2), which stems from CPU-GPU synchronization in the bisection solver.
>
> End-to-end per-step latency for 4M tokens/step on NVIDIA B200.
>
> | | AdamW | Muon | MuonSphere | Spectral Sphere |
> |---|---|---|---|---|
> | Time (ms) | 6734.15 (-2.10%) | 6878.83 | 6949.85 (+1.03%) | 7666.32 (+11.45%) |
>
> However, the solver overhead is a fixed per-step cost independent of batch size. As batch size grows, forward/backward computation dominates, so SSO's step-wise advantage translates more directly into wall-clock gains at scale. This overhead can be further reduced by GPU-native solvers (Appendix A.6).
>
> **Q4: Transfer of $c$ and module granularity guidance**
>
>  Yes, the optimal $c$ transfers perfectly across architecture sizes. Regarding why $c=2$ is often the optimal choice, we have a theoretical conjecture.
> For a weight matrix $W \in \mathbb{R}^{d_{out} \times d_{in}}$ with Kaiming initialization $\sigma^2 = 1/d_{in}$, random matrix theory states that its spectral norm is approximately $\sigma (\sqrt{d_{in}} + \sqrt{d_{out}})$.
>
> In our framework, we constrain the spectral norm to be $c\sqrt{d_{out}/d_{in}}$. Equating these two expressions yields:
>
> $$
> c \sqrt{\frac{d_{out}}{d_{in}}} = \frac{1}{\sqrt{d_{in}}} (\sqrt{d_{in}} + \sqrt{d_{out}}) \implies c = 1 + \sqrt{\frac{d_{in}}{d_{out}}}
> $$
>
> For square matrices (where $d_{in} = d_{out}$), this gives exactly $c = 2$. In standard Transformers, most projections have $d_{in}/d_{out}\approx 1$, so the ideal $c$ naturally hovers around 2.  While per-module tuning is theoretically possible, it introduces considerable  complexity, so we adopt a single unified $c$ across modules.
>
> **Module Granularity:** In principle, we should shard modules according to their distinct functional roles. However, splitting into overly small matrices is detrimental to GEMM performance. We conducted detailed ablations and found that, for example, in most cases, the FFN up/gate projections need not be split—whether to shard them or not has negligible impact on downstream evaluation performance. We can keep them fused to achieve better infrastructure efficiency.
>
>  **Q5: Figure 11 — AdamW parametrization**
>
> Our AdamW baseline also uses the same $\mu$P initialization and learning rate scaler. In principle, $\mu$P only guarantees that a constant learning rate yields optimal feature learning in the infinite-width limit; it does not provide theoretical guarantees for practical finite-width settings (where "infinite-width" refers to increasing model width until loss no longer decreases).
>
> Several works have noted this phenomenon that mup adamw may still drifts in optimal lr . For example, "Weight Decay may matter more than $\mu$P for Learning Rate Transfer in Practice" (https://arxiv.org/abs/2510.19093) attempts to adjust weight decay to achieve better lr transfer (see figure1, the drift is also significant) . Our SSO demonstrates better curve overlap compared to both Adam and Muon (satisfying $\mu$P perfectly in both forward and backward passes).
>
> **Presentation (Figure Legibility)**
>
> Thank you for the suggestion. We will enlarge the axis labels, legends, and overall font sizes in Figures 1, 3, and others to significantly improve readability in the final version.

---

> > ### Author Rebuttal · Reviewer_xbZJ · 2026-04-01
> >
> > Thank you to the authors for their additional clarifications which were helpful. Given the high quality of both the work and the response, I will maintain my initial score.

---

### Official Review · Reviewer_kUBU · 2026-03-14

**Soundness:** 4
**Presentation:** 4
**Significance:** 4
**Originality:** 3
**Overall Recommendation:** 5
**Confidence:** 5

**Summary:**

See below

**Compliance With Llm Reviewing Policy:**

Affirmed.

**Final Justification:**

I have carefully read the rebuttal by authors. I will keep my score and vote for clear acceptance.

**Key Questions For Authors:**

**Main Contributions:**

The paper introduces a new method for training LLMs by constraining the optimization process on a spectral sphere.  The paper studies the steepest descent method under the constraints, and then provide a proposal to implement the steepest descent method efficiently.



**Strengths:**

The topic is timely and important, as training stability for LLMs remains a critical challenge. The paper is beautifully written. The presentation is of high standard and the paper is enjoyable to read. As a reviewer, it is easy to tell that the authors spent great effort polishing the presentation of the paper.

The paper attempts to bridge theoretical insights with practical implementation, which is commendable.  The experiments are thorough, and  the experiments on 200-layer Transformer is intriguing.   The proposed method is theoretically elegant, and the empirical validation seems sound and solid.

Overally speaking, this is a high-quality paper, and the paper has some interesting design that is worth sharing with the community.  I vote for clear acceptance.



**Questions:**

1. Does the method require tuning additional hyperparameters like lr or batch size? If so, how should they be set? Any suggestions?
2. Any insight on the choice of the radius c?
3. Please correct me if I missed something: I cannot seem to find the optimal lambda reported in the script. What is usually the optimal dual variable lambda found by the bisection procedure?  Is the optimal close to 0?  If true, what is the implication?



**Typos and Presentation:**

I carefully go through the paper and find the following typos:

1. Intro line 73: `sharding\citep` → `sharding \citep`
2. line 860: `weight matrics` → `weight matrices`
3. line 729: `Abaltion` → `Ablation`
4. line 619: `Singular Vector Decomposition` → `Singular Value Decomposition`
5. line 44: `activations remains` → `activations remain`
6. line 307: `This heuristic effectively balancing` → `This heuristic effectively balances`
7. line 318: `we use` → `We use`
8. line 730: `instable` → `unstable`
9. line 713: `$f(\lambda)$ is indeed monotonic` → `$h(\lambda)$ is indeed monotonic`

**Limitations:**

See above.

**Strengths And Weaknesses:**

See below

---

> ### Author Rebuttal · Authors · 2026-03-30
>
> Thank you for the careful reading and the detailed list of typos. We address your questions below.
>
> ### **Q1: Does SSO require tuning lr or batch size?**
>
> No special tuning is required for the batch size or the learning rate. For the learning rate, SSO follows the standard $\mu$P transfer protocol: we sweep for the optimal learning rate on a small proxy model and directly transfer it to larger widths (as shown in Fig 2). Because of this property, **tuning the learning rate is actually easier and more predictable** than standard parameterizations. For the batch size and momentum coefficient $\beta$, we use the exact same configurations as Muon.
>
> ### **Q2: Choice of radius $c$**
>
> The radius $c$ explicitly controls the branch output magnitude relative to the residual stream (as discussed in Section 4.1). Regarding why $c=2$ is often the optimal choice, we have a theoretical conjecture rooted in random matrix theory and standard initialization schemes.
>
> For a weight matrix $W \in \mathbb{R}^{d_{out} \times d_{in}}$ with kaiming init $\sigma^2 = 1/d_{in}$, random matrix theory states that its spectral norm is approximately $\sigma (\sqrt{d_{in}} + \sqrt{d_{out}})$.
>
> In our framework, we constrain the spectral norm to be $c\sqrt{d_{out}/d_{in}}$. Equating these two expressions yields:
>
> $$
> c \sqrt{\frac{d_{out}}{d_{in}}} = \frac{1}{\sqrt{d_{in}}} (\sqrt{d_{in}} + \sqrt{d_{out}}) \implies c = 1 + \sqrt{\frac{d_{in}}{d_{out}}}
> $$
>
> For square matrices (where $d_{in} = d_{out}$), this gives exactly $c = 2$. In standard Transformer architectures, the ratio $d_{in}/d_{out}$ for most linear projections is typically 1, meaning the ideal $c$ for each layer naturally hovers around 2, albeit with slight variations. While it is theoretically possible to assign a distinct, tunable scale $c$ to each individual module, tuning these layer-wise coefficients is computationally prohibitive and risks introducing complex couplings with other hyperparameters. Therefore, we adopt a single, unified $c$ across modules.
>
> ### **Q3: Typical value of $\lambda^\star$ and its implications**
>
> In practice, the optimal dual variable $\lambda^\star$ is close to 0 and fluctuates around it in most steps (which is consistent with the distribution shown in Figure 3).
>
> The implication of $\lambda^\star \approx 0$ is that the tangent space correction applied to the `msign` direction is small in magnitude. This is theoretically expected, likely because high-dimensional random vectors tend to be approximately orthogonal, making the unconstrained update already somewhat close to the tangent space. However, this small correction is *persistent*. Its cumulative effect over thousands of training steps is exactly what separates our exact SSO from the MuonSphere baseline (which corresponds to strictly setting $\lambda=0$ without the steepest descent correction). This persistent accumulation translates to the consistent empirical performance gap we observe in Table 3.
>
> ### **Typos and Presentation**
>
> We have confirmed all 9 typographical errors you pointed out and will fix them in the revision. In particular, "Singular Vector Decomposition" $\to$ "Singular Value Decomposition" and "$f(\lambda)$" $\to$ "$h(\lambda)$" were indeed oversights on our part. We deeply appreciate your meticulous review and your help in improving the polish of our paper!

---

> > ### Author Rebuttal · Reviewer_kUBU · 2026-04-01
> >
> > I would like to thank the authors for the detailed rebuttal. I will keep my score and vote for clear acceptance.

---

### Official Review · Reviewer_ussv · 2026-03-18

**Soundness:** 3
**Presentation:** 3
**Significance:** 3
**Originality:** 2
**Overall Recommendation:** 4
**Confidence:** 3

**Summary:**

The paper identifies a limitation of current optimization strategies in large language models that methods like Muon efficiently control spectral norms of weight updates, but fail to constrain the weights themselves, so they are not remaining $\Theta(1)$. To address this issue, the authors introduce Spectral Sphere Optimizer (SSO) to go beyond the Maximal Update Parametrization ($\mu P$) principles. To fix the problem, SSO strictly enforces module-wise spectral constraints on both weights *and* updates by finding the steepest descent direction within the spectral sphere's tangent space, followed by a retraction step. To make this practical at scale, the authors implement system-level optimizations in Megatron, including atomic module sharding and an adaptive kernel dispatcher. Empirically, SSO demonstrates improved convergence, enhanced MoE router load balancing, and strictly bounded activations across Dense 1.7B, MoE 8B, and 200-layer DeepNet architectures compared to AdamW and Muon.

**Compliance With Llm Reviewing Policy:**

Affirmed.

**Final Justification:**

I updated my score as the authors answered my questions and responded to my concerns.

**Key Questions For Authors:**

1. Why not just normalize the matrices by their norm instead of working with top singular components?

2. There is a certain disconnect between the message the authors communicate through Figure 1 (a), and the spectral norm considerations given in Section 2. In particular, Figure 1 talks about the maximal activation, while the authors base their derivation on the analysis of RMS norms. While Figure 5 (b), shows a change in AbsMax of FFN activations as well, why not tuckle AbsMax directly instead of RMS norms?

3. The learning rate is annealed to 10% peak, which is still pretty high. Is it because the training budget is limited (undertrained model) and annealing to a smaller number is not required?

**Limitations:**

yes

**Strengths And Weaknesses:**

Strength 1. The $\mu P$ fixes are important: The unification of steepest descent with strict $\mu P$ constraints is well-reasoned, directly targeting the root causes of activation drift and feature instability over long training horizons. This may have significant impact on practical pipelines of training large models.

Strength 2: Practical Systems Engineering: The authors provide a comprehensive infrastructure design (atomic sharding, ping-pong load balancing, multi-stream execution) to mitigate the inherent computational bottlenecks of the required root-finding solver.

Strength 3: Solid Empirical Validation: The evaluations across diverse architectures (MoE, ultra-deep 200-layer networks) effectively highlight SSO's benefits, particularly in improving MoE routing stability and handling extreme depth. The time tests were run on Nvidia B200, which makes them even more relevant.

Weakness 1: Heuristic Hyperparameters: The target spectral radius $R$ relies heavily on a scaling scalar parameter $c$, which requires manual tuning to balance the residual stream's signal-to-noise ratio.

Weakness 2: Redundancy in Projection Logic: The norm constrained on the weights is maintained only approximately when doing a first-order approximation of the bound. The authors address this by projecting the weights before each step (retraction), but then it's not clear why not use this approach in the first place. Instead of using the top singular vectors of $W$ to preserve the spectral norm, one could simply divide $W$ by its singular norm at each iteration. Since everything is an approximation and we're not presented with evidence the proposed approach is the right one, I seriously doubt the theoretical foundation of the paper.

Weakness 3: Computational Overhead: Despite the system-level optimizations, the bisection solver for the Lagrange multiplier still introduces non-trivial latency and synchronization overhead between the CPU and GPU.

Weakness 4: I don't particularly like the arguments written in Section 3.2, where the authors exclude the consideration of matrices with more than one largest singular value by saying their measure is 0. While the measure statement itself is true, we're not dealing with randomly generated matrices, they are obtained through a training process, and therefore, I'd have preferred if the authors were more careful with their arguments. As far as I can see, the only difference would be that $\Theta$ is the sum of all pairs of singular vector outer products.

### Minor

"Yang et al. (2024) shows that" -> "Yang et al. (2024) show that".
What's the point of writing the equation in lines 218-219? There is no discussion around it whatsoever.
Line 521: Did you mean "both $W$ and $W+\Delta W$" instead of "both $W$ and $\Delta W$"?.

> We therefore eliminate weight decayin hidden 2D weights, removing a sensitive hyperparameter from training.

I think this claim is exaggerated since the authors introduce a different hyperparameter. And while weight decay is know to work on most problems with the value of 0.1, it's not clear how universal is the new hyperparameter $c$ due to limited testing.

In Figure 9, the authors claim that
> AdamW shows pronounced instability, characterized by frequent loss spikes

there are only 3 loss spikes in total, and they don't seem to hurt the model performance as the model recovers almost immediately. I think the claims surrounding this loss curve are exaggarated.

---

> ### Author Rebuttal · Authors · 2026-03-30
>
> We would like to thank the reviewer for the careful evaluation of our paper! We greatly appreciate your insightful comments and suggestions. We address your concerns point-by-point below:
>
> **W2 & KQ1: Why not just normalize by the spectral norm?**
> The approach you describe — updating freely then dividing by the spectral norm — is precisely our MuonSphere baseline (Section 3.3). The key difference is that SSO solves for the steepest descent direction *within* the tangent space (Eq. 10), whereas MuonSphere updates within the tangent space but does not follow the steepest descent direction. The empirical gap between SSO and MuonSphere (Fig 7, Table 3: 56.35 vs 56.19 Avg Acc) confirms this distinction is not merely theoretical but practically significant. We will make this comparison more explicit in the revision.
>
> **W4: Measure-zero argument for singular value multiplicity**
> Thank you for pointing this out. We agree that the original measure-zero argument, phrased in terms of random matrices, was not the cleanest justification for weights produced by optimization. We will revise Section 3.2 accordingly. Our derivation does not fundamentally rely on randomness: when the top singular value is simple, $\nabla_W \|W\|_2 = u_1 v_1^T$; when it is not simple, the correct replacement is any subgradient $\Theta \in \partial\|W\|_2$ rather than a unique gradient. Hence the degenerate case does not invalidate the argument; it only replaces a singleton gradient with a set-valued subdifferential. We will make this explicit and remove the random-matrix phrasing.
>
> **W3: Computational overhead**
> The bisection solver introduces a fixed per-step cost primarily due to CPU-GPU synchronization. As batch size grows, this constant overhead is effectively amortized. Furthermore, a GPU-native solver (as discussed in Appendix A.6) will eliminate the synchronization latency and further close the computational gap. We will clarify this optimization trajectory in the final version.
>
> **W1 & Minor: Weight decay vs $c$**
> We will soften the claim regarding the elimination of hyperparameters. That said, the hyperparameter $c$ differs fundamentally from weight decay. $c$ *explicitly and strictly* determines the spectral norm of the weight matrix. In contrast, weight decay only *implicitly* regularizes the norm, and the resulting steady-state norm heavily depends on the complex interplay of training dynamics, learning rate, and gradient variance. Thus, $c$ offers a more direct and predictable control over the network's Lipschitz constant.
>
> **KQ2: AbsMax vs RMS**
> The $\mu$P theory and our theoretical framework bound RMS norms, which is why our derivation targets RMS. Figure 1(a) shows AbsMax to highlight an important empirical observation: bounding the RMS norm effectively suppresses extreme outlier activations (AbsMax) as well. We will clarify this connection in the text so there is no disconnect between the theory and the figure.
>
> **KQ3: LR annealing to 10%**
> Annealing the learning rate to 10% of its peak is a common practice in modern LLM pre-training, rather than a symptom of an undertrained model. For instance, Deepseek-v3 (https://arxiv.org/abs/2412.19437) and Kimi-k2 (https://arxiv.org/abs/2507.20534) both explicitly decay their learning rates to 10% of the maximum value. Our schedule simply follows this well-established community standard. Also, our model is 100B tokens over 1.7B model, the D/N ratio is 100/1.7=58, about 2.9x chinchilla optimal scaling law (https://arxiv.org/abs/2203.15556), proving that model training is sufficient.
>
> **Minor comments**
> - We will fix all typos, including "Yang et al. (2024) show that".
> - Regarding the phrasing "both $W$ and $\Delta W$", we actually intended to write "both $W$ and $W+\Delta W$". We will fix this typo.
> - The equation on the Frobenius norm bound (lines 218–219, after Eq. 13) shows that bounding $\|W\|_2$ automatically bounds $\|W\|_F$, which provides further theoretical justification for removing weight decay. We will add the missing discussion around this equation.
> - We will soften the claim about AdamW's instability in Figure 9.
>
> We hope these clarifications address your concerns.

---

> > ### Author Rebuttal · Reviewer_ussv · 2026-04-01
> >
> > I thank the authors for their response, I'll raise my score as I found the answers to be quite reasonable.

---

### Decision · Program_Chairs · 2026-04-30

**Decision:**

Accept (spotlight)

**Comment:**

This paper proposed SSO towards fully aligning muP, in order to achieve better loss convergence and learning rate transfer. The authors achieve this by both constraining the weights and weight update on spheres. The experiments are very promising and diverse. Additional efforts are put into efficient implementation on Megatron, rendering it applicable to large-scale adoption. The reviewers have some concerns about fair comparison and hyperparameter settings, which are sufficiently addressed during the rebuttal. However, the authors should add the rebuttal and implement the changes in writing.